# Adiponectin, a Therapeutic Target for Obesity, Diabetes, and Endothelial Dysfunction

**DOI:** 10.3390/ijms18061321

**Published:** 2017-06-21

**Authors:** Arunkumar E. Achari, Sushil K. Jain

**Affiliations:** Department of Pediatrics, Louisiana State University Health Sciences Center, 1501 Kings Highway, Shreveport, LA 71103, USA; aeluma@lsuhsc.edu

**Keywords:** adiponectin, obesity, type 2 diabetes, endothelial dysfunction

## Abstract

Adiponectin is the most abundant peptide secreted by adipocytes, whose reduction plays a central role in obesity-related diseases, including insulin resistance/type 2 diabetes and cardiovascular disease. In addition to adipocytes, other cell types, such as skeletal and cardiac myocytes and endothelial cells, can also produce this adipocytokine. Adiponectin effects are mediated by adiponectin receptors, which occur as two isoforms (AdipoR1 and AdipoR2). Adiponectin has direct actions in liver, skeletal muscle, and the vasculature.Adiponectin exists in the circulation as varying molecular weight forms, produced by multimerization. Several endoplasmic reticulum ER-associated proteins, including ER oxidoreductase 1-α (Ero1-α), ER resident protein 44 (ERp44), disulfide-bond A oxidoreductase-like protein (DsbA-L), and glucose-regulated protein 94 (GPR94), have recently been found to be involved in the assembly and secretion of higher-order adiponectin complexes. Recent data indicate that the high-molecular weight (HMW) complexes have the predominant action in metabolic tissues. Studies have shown that adiponectin administration in humans and rodents has insulin-sensitizing, anti-atherogenic, and anti-inflammatory effects, and, in certain settings, also decreases body weight. Therefore, adiponectin replacement therapy in humans may suggest potential versatile therapeutic targets in the treatment of obesity, insulin resistance/type 2 diabetes, and atherosclerosis. The current knowledge on regulation and function of adiponectin in obesity, insulin resistance, and cardiovascular disease is summarized in this review.

## 1. Introduction

Life style modification and rapid urbanization has triggered the obesity epidemic, which is associated with a number of health problems. Obesity is often summarized together as the metabolic syndrome and increases the risk of insulin resistance, type 2 diabetes, fatty liver disease and cardiovascular disease [1]. Adiponectin, also known as adipocyte complement-related protein of 30 kDa (*Acrp30*), a type of adipokine, was identified by different groups [2,3,4]. Adiponectin is an endocrine factor synthesized and released from adipose tissue [5]. Basic scientific studies have demonstrated that adiponectin has insulin-sensitizing [5], anti-atherogenic, and anti-inflammatory properties [6,7]. Therefore, it is important for investigators to have a thorough understanding of adiponectin. Such knowledge may lead to new therapeutic approaches for diseases, such as type 2 diabetes, metabolic syndrome, cardiovascular disease, and obesity.

## 2. Adipose Tissue Biology: A Brief Overviews

Adipose tissue, commonly called “fat”, is a type of loose connective tissue comprised of lipid-filled cells (adipocytes) surrounded by a matrix of collagen fibers, blood vessels, fibroblasts, and immune cells [8]. Two adipose tissues with different functions coexist in humans: white adipose tissue (WAT) and brown adipose tissue (BAT). WAT represents the vast majority of adipose tissue in the organism. Adipose tissue stored energy in the form of triglycerides and cholesterol as a single large lipid droplet (unilocular appearance), whereas brown adipose tissue is involved in nonshivering thermogenesis as a result of fat burning. Brown adipocytes are particularly present in small mammals and human neonates, and contain several smaller lipid droplets (multilocular appearance) [9]. WAT is composed of many cell types, adipocyte being the most abundant. The other cells, collectively referred to as the stromalvascular fraction (SVF), are a heterogeneous population of endothelial cells, macrophages, fibroblasts, stem cells, and lymphocytes [10].

## 3. Adiponectin: Biosynthesis, Structure and Downstream Signaling

Adiponectin (also known as *Acrp30* [2], *AdipoQ* [11], *GBP-28* [12], and *apM1* [3]) is a 244-amino acid protein secreted mainly by the adipose tissue. It was identified almost simultaneously by fourdifferent groups using different approaches [2,3,11,12]. Initially, it was thought that adiponectin was exclusively produced only by adipose tissue. However, later it has proven, from different research groups, that adiponectin is expressed in other tissues including human and murine osteoblasts [13], liver parenchyma cells [14], myocytes [15], epithelial cells [16], and placental tissue [17]. Human adiponectin is encoded by the *Adipo Q* gene, which spans 17 kb on chromosome locus 3q27. The gene for human adiponectin contains three exons, with the start codon in exon 2 and stop codon in exon 3 [18,19]. This human chromosome 3q27 has been identified as a region carrying a susceptibility gene for type 2 diabetes and metabolic syndrome [20,21]. Serum levels of adiponectin decrease with obesity and are positively associated with insulin sensitivity [22,23]. Because of these positive actions, adiponectin has attracted tremendous scientific interest in recent years, and has been extensively studied both in human and animal models.

### 3.1. Structural Features, Synthesis and Post Translational Modification of Adiponectin

Adiponectin is a 30 kDa multimeric protein and is secreted mainly by white adipose tissue, although other tissues express low levels of adiponectin too. Full-length human adiponectin comprises 244 amino acid residues, including a NH_2_-terminal hyper-variable region (amino acids from 1–18), followed by a collagenous domain consisting of 22 Gly-XY repeats, and a COOH-terminal C1q-like globular domain (amino acids from 108–244). In contrast to humans, mouse adiponectin is a 247 amino acid long protein [2]. Adiponectin is secreted from adipocytes into the bloodstream as three oligomeric complexes, including trimer (67 kDa), hexamer (140 kDa), and a high molecular weight (300 kDa) multimer comprising of at least 18 monomers (Figure 1). The monomeric form of adiponectin is undetectable in native conditions. Homotrimer, also known as low molecular weight (LMW), is a basic building block of oligomeric adiponectin. The interaction between the collagenous domains results in formation of highly ordered trimer, which is further stabilized by an intratrimer disulfide bond mediated by Cys^39^ (or Cys^22^, if the N-terminal 17-amino acid secretory peptide is excluded). The formation of a disulfide bond between two trimers mediated by the free Cys^39^ in each leads to the formation of the hexameric form of adiponectin. This hexameric form serving as the building block for the HMM form, which consists of 12–18 hexamers existing in a bouquet-like structure [24]. Post-translational modifications, especially hydroxylation and subsequent glycosylation of several highly conserved lysine residues within its collagenous domain, are crucial for the formation of HMW oligomeric adiponectin, which is the major bioactive isoform contributing to its insulin-sensitizing and cardiovascular protective effects [25]. Globular adiponectin, the globular C1q domain of adiponectin generated from full-length protein by proteolysis, is also biologically active [26].

The biosynthesis and secretion of adiponectin in adipocytes are tightly controlled by several molecular chaperones in the endoplasmic reticulum, including: ERp44 (Endoplasmic Reticulum resident protein 44), Ero1-La (ER oxidoreductase 1-La), and DsbA-L (disulfide-bond A oxidoreductase-like protein) [27,28,29]. Scherer and his colleagues proved that ERp44 retains adiponectin oligomers in the endoplasmic reticulum via a thiol-mediated mechanism [2]. ERp44 forms a mixed disulfide bond with adiponectin through the cysteine residue within its variable region (Cys^36^ in humans, and Cys^39^ in mice) [27]. In contrast with the inhibitory effects of ERp44, Ero1-Lα selectively enhances the secretion of HMW adiponectin. Ero1-Lα can displace the ERp44-retained HMW adiponectin, and, therefore, release this oligomeric complex trapped by ERp44 [27]. DsbA-L functions as a protein disulfide isomerase to regulate adiponectin disulfide bond formation, which is essential for multimerization. Sialic acids also modified adiponectin through O-linked glycosylation situated on threonine residues within the hypervariable region [22], which determines the half-life of adiponectin in the circulation by modulating its clearance from the bloodstream. In addition, succination of the highly conserved cysteine residues (Cys^36^) within the hypervariable region of adiponectin blocks adiponectin multimerization, and it may contribute to the decrease in plasma adiponectin in diabetes [30]. Therefore, extensive post-translational modifications of adiponectin are essential for efficient maturation, oligomerization, and secretion of adiponectin, and are also important for maintaining its stability in the circulation.

### 3.2. Adiponectin Receptors

AdipoR1 and AdipoR2, two structurally related seven transmembrane receptors, have been identified to function as adiponectin receptors. They are structurally and functionally distinct from classical G-protein coupled receptors (GPCR). Unlike all other GPCRs reported, AdipoR1 and AdipoR2 have an inverted membrane topology with a cytoplasmic NH_2_ terminus and a short, extracellular COOH terminal domain of approximately 25 amino acids [31]. AdipoR1 and AdipoR2 encoded by genes situated on the 1p36.13-q41 and 12p13.31 chromosomal regions, respectively [31]. AdipoR1 is a high affinity receptor for globular adiponectin and a low affinity receptor for full length adiponectin. It is expressed ubiquitously, but most abundantly, in skeletal muscle. On the other hand, AdipoR2 mainly recognizes full length adiponectin and is predominantly expressed in the liver [31].

Apart from AdipoR1 and AdipoR2, another receptor has also been identified for adiponectin, called T-cadherin. It acts as a receptor for hexameric and HMW forms of adiponectin but not for other forms [32]. T-cadherin is a unique cadherin molecule that is anchored to the surface membrane, not through a transmembrane domain, but, instead, via a glycosyl phosphatidyl inositol (GPI) moiety [33,34,35]. Recent studies have identified that cadherin-deficient mice showed elevated plasma adiponectin levels, especially HMW form [34].

### 3.3. Role of APPL1 and APPL2 in Adiponectin Signaling

APPL1, an adaptor protein, binds to the adiponectin receptors and positively mediates adiponectin signaling in mammals. APPL1 has three functional domains, which play an important role in the intracellular signal transduction of adiponectin receptors pathway. This includes NH_2_-terminal Bin1/amphiphysin/rvs167 (BAR) domain (initially identified as the leucine zipper motif, 18–226 amino acids), followed by a pleckstrin homology (PH) domain (278–377 amino acids) and a phosphotyrosine binding (PTB) domain (597–636 amino acids) near the COOH terminus [36]. APPL1 acts as an interacting partner of both AdipoR1 and AdipoR2. Broadly, the BAR domain is associated with multiple biological processes like sensing and inducing membrane curvature, small GTPase binding [37,38,39] transcriptional repression, apoptosis, and secretory vesicle fusion [40,41], and it is located near to the NH_2_ terminus. In general, the PH domain targets proteins to specific membrane compartments by increasing the lipid specificity of the BAR domain [42]. The general function of PTB domain is to act as an adaptor or scaffold for the binding of proteins. The PTB domain of APPL1 is located near the COOH terminus, away from BAR-PH domain, making it an easily accessible structure for its binding partners. APPL2 is an isoform of APPL1, and these two proteins display 54% identity in protein sequences [43]. Similar to APPL1, APPL2 has an N-terminal BAR domain, central PH domain, and C-terminal PTB domain. APPL2 involves follicle-stimulating hormone signal transduction pathway by binding to APPL1 via their respective BAR domains [44].

### 3.4. Downstream Signaling Events of Adiponectin

Adiponectin elicits a number of downstream signaling events. APPL1 acts as a signaling pathway mediator in cross-talk with adiponectin and insulin, and it interacts directly with insulin receptor substrates. Activation of insulin receptor substrate proteins serve as docking platforms for the p85 regulatory subunit of the phosphatidylinositol 3-kinase (PI3K), which results in the generation of phosphatidylinositol 3,4,5-triphosphate at the plasma membrane. This activation of the PI3K pathway activates Akt and its downstream targets, which in turn exhibits a biological response [45]. Reports from various research groups demonstrate that APPL1 is involved in the activation of AMP activated protein kinase (AMPK) [46,47]. Upon binding of adiponectin to its receptor, APPL1 binds and activates protein phosphatase 2A, resulting in dephosphorylation and inactivation of protein kinase Cz (PKCz). This in turn dephosphorylates liver kinase B1 (LKB1) at its Ser^307^, allowing LKB1 to translocate from nucleus to cytoplasm, and activate AMPK [46]. Activation of AMPK is a key step in mediating the most of the effects of adiponectin at cellular level. AMPK, is a fuel-sensing enzyme that responds to decreases in cellular energy state by activating pathways that generate energy (e.g., oxidation of fats), and inhibiting energy consuming pathways; however, AMPK is not acutely necessary for survival (e.g., fatty acid, triglyceride, and protein synthesis). Adiponectin drastically increases the expression and activity of PPAR-α, a key transcription factor in metabolic regulation, which in turn upregulates acetyl CoA oxidase (ACO) and uncoupling proteins (UCPs); thereby, promoting fatty acid oxidation and energy expenditure [31]. Interestingly, the action of APPL1 by adiponectin on p38 MAPK [48] and Rab5 a GTPase downstream of APPL1, improves glucose metabolism in various metabolic tissues [36]. Activated AMPK, in response to adiponectin, is also involved in nitric oxide production through the activation of eNOS, resulted in vasodilation [49]. In addition, activated AMPK by adiponectin inhibits IKK/NFκB/PTEN triggered apoptosis [50] (Figure 2).

### 3.5. Interactions between the AMPK and Insulin Signaling Pathways

The insulin signaling pathway is activated when nutrients are available, whereas the AMPK pathway is activated when cells are starved for a carbon source. One would therefore expect these 2 pathways to oppose each other, and this is often the case. In mammals, insulin promotes lipid, protein, and glycogen synthesis, whereas AMPK inhibits these biosynthetic pathways. Proteiogenic effect of insulin is mediated in part by activation of the target of rapamycin TOR pathway via phosphorylation of TSC2, whereas AMPK activation causes phosphorylation of different sites on TSC2 and inhibits TOR [51,52]. In some tissues, such as heart, insulin inhibits AMPK and this action was mediated by the activation of the protein kinase B. In other cases, the insulin and AMPK signaling pathways work in the same direction, particularly in processes that regulate plasma glucose levels. In skeletal muscle, both insulin and AMPK activation promotes glucose uptake by increasing GLUT4 translocation to the plasma membrane. However, the fate of the glucose is different in either case. In the case of insulin, glucose can be stored as glycogen (anabolic), whereas, in the case of AMPK, glucose can be entered in oxidative (catabolic) pathway. These two pathways appear to converge on the phosphorylation of AS160, which further involved in GLUT-4 translocation [53,54]. A second case in which insulin and AMPK act in the same direction occurs in the liver, in which both repress the expression of enzymes of gluconeogenesis, such as phosphoenolpyruvate carboxykinase and glucose-6-phosphatase [55]. It makes sense that insulin, a hormone released in response to high blood glucose, should repress hepatic glucose production, whereas in the case of AMPK it may perhaps have evolved as among its anti-anabolic actions.

## 4. Adiponectin Signaling in Key Metabolic Tissues

Adiponectin exhibits anti-diabetic, anti-inflammatory, and anti-atherogenic effects, and it also functions as an insulin sensitizer. Hence, it is a novel therapeutic target for diabetes and metabolic syndrome [6]. Adiponectin also plays a central role in energy homeostasis through its action in hypothalamus, and a new role for adiponectin as a “starvation gene” has been proposed [56,57]. The following section discusses the signal transduction of adiponectin in different tissues and the role of APPL1 in mediating the effects of adiponectin.

### 4.1. Skeletal Muscle

Many studies have clearly demonstrated that the skeletal muscle is an important peripheral target tissue for adiponectin to exert its beneficial metabolic effects. Previous work reported that adiponectin improved glucose utilization and fatty acid oxidation in C2C12 myocytes [58]. In addition, in mice fed with high fat/sucrose diet, adiponectin showed to increase energy expenditure by increasing fatty acid oxidation, and to increases glucose uptake in skeletal muscle [26]. Studies with rat skeletal muscle cells have shown that globular adiponectin increases glucose transporter-4 (GLUT-4) translocation and glucose uptake [59]. AdipoR1 is the most abundant adiponectin receptor in skeletal muscle, and globular adiponectin showed high affinity towards AdipoR1. By this reason, most of the adiponectin effects in skeletal muscle are carried out by globular form [31].

The binding of adiponectin to its membrane receptors, such as AdipoR1 and AdipoR2, leads to the activation of two major signal pathways in muscle cells, the AMPK and the p38 mitogen-activated protein kinase (MAPK) pathways [31,43]. Activation of these pathways has been shown to be essential for adiponectin-induced glucose uptake and fatty acid oxidation. Globular and full length adiponectin was found to increase phosphorylation and activation of AMPK; thereby, it increases fat oxidation and glucose uptake in C2C12 myoctyes [58]. Similar to previous studies, mice injected with globular adiponectin showed increased expression of molecules involved in fatty-acid transport, combustion, and energy dissipation, such as such as CD36, ACO and UCP2. This turn leads to decrease in tissue triglyceride content in mice skeletal muscle [5].

Recently it has been shown that cellular ceramide levels were lowered by adiponectin through the activation of ceramidase, which inturn converts ceramide to sphingosine. This effect appears to be dependent on activation of AdipoR1 or AdipoR2 [60]. In addition, overexpression of adiponectin, AdipoR1, or AdipoR2 in liver reduces hepatic ceramide levels and improves insulin sensitivity, while deficiency of adiponectin increases hepatic ceramide levels and exacerbates insulin resistance. Since accumulation of ceramide in skeletal muscle has been reported to be associated with impaired insulin sensitivity [61], decreased ceramide concentration by adiponectin may be a mechanism for increasing insulin sensitivity in skeletal muscles. In addition, it has been suggested that adiponectin decreases insulin resistance by decreasing muscular lipid content in obese mice [5].

p38 MAPK, which belongs to the group of stress-activated kinases, has been connected with a large number of cellular processes, including cell growth and differentiation, apoptosis, and inflammation; in addition, it gets activated in response to a numerous extracellular stimuli [62]. Previous studies have suggested that the activation of p38 MAPK and PPAR-α along with AMPK by globular adiponectin was found to induce fatty acid oxidation in C2C12 skeletal muscle cells [63]. Similarly, p38 MAPK acts downstream of AMPK in cardiomyocytes, and the inhibition of the AMPK/p38 MAPK signaling pathway partially abolishes the stimulation of glucose uptake in response to hypoxia [64]. It has recently been reported that adiponectin enhances fatty acid oxidation in muscles cells by stimulating PPARα transcriptional activity via sequential activation of AMPK and p38 MAPK [63].These results suggest the presence of multiple pathways that could mediate the effect of adiponectin on glucose and the fatty acid metabolism in muscles.

From the various studies, it was evident that APPL1 plays an important role in adiponectin signaling. In cultured skeletal muscle cells, overexpression of APPL1 enhances the phosphorylation and activation of AMPK and p38 MAPK. On the other hand, suppression of APPL1 inhibited adiponectin meditated AMPK as well as p38 MAPK and ACC phosphorylation, and, thereby, limited fat oxidation in C2C12 myotubes, implies APPL1 plays a crucial role in adiponectin signaling in the skeletal muscle [43].

Adiponectin reduces plasma glucose levels in mice subjected to high fat meal [26], and directly regulates glucose metabolism and insulin sensitivity in C57BL6J mice [58]. This beneficial effect of adiponectin on glucose metabolism was mainly via the activation of AMPK [58], which leads to the translocation of GLUT4 to the cell membrane. Both forms of adiponectin upregulates GLUT-4 membrane translocation in rat skeletal muscle cells, and the adaptor protein, APPL1, acts as the first signaling molecule that binds to the adiponectin receptors and positively mediates adiponectin signaling in muscle cells [43]. Further, overexpression of APPL1 increases, and suppression of APPL1 level reduces its action. Thus, adiponectin signaling and adiponectin-mediated downstream events in mouse skeletal muscle cells include: lipid oxidation, glucose uptake, and the GLUT-4 membrane translocation [43]. Therefore, binding of adiponectin with APPL1 mediates phosphorylation of AMPK and p38 MAPK, thereby adiponectin stimulates GLUT-4 translocation in muscle cells. In addition, recently small GTPase, Rab5, has been shown to interact with APPL1 and to regulate GLUT4 internalization in an insulin-dependent mechanism [43].

### 4.2. Vascular Endothelium

Atherosclerosis is the process of vascular wall thickening and hardening, and it is the primary cause of coronary heart disease, ischemic stroke, and peripheral arterial disease [65]. Numerous epidemiological studies suggest that adiponectin deficiency (hypoadiponectinemia) is associated with coronary artery disease and hypertension [66], left ventricular hypertrophy [67], and a greater risk of myocardial infarction [68]. Experimental studies with cell cultures and animal models have shown cardioprotective action of adiponectin in cell types, including: vascular endothelial cells, smooth muscle cells, and cardiac myocytes and adiponectin-deficient mice [7]. The vasculoprotective and angiogenic properties of adiponectin have demonstrated in adiponectin-deficient mice in which adiponectin improves revascularization of ischemic limbs [69] and rescues from cerebral ischemia-reperfusion [70]. Additionally, adiponectin supplementation attenuates neointimal thickening in mechanically injured arteries through the suppressive action of adiponectin on the proliferation and migration of vascular smooth muscle cells [7]. On the high salt diet, adiponectin knock-out mice developed severe blood pressure due in part to a reduction of endothelial nitric oxide synthase activity [71]. In addition, studies had shown that overexpression of adiponectin protects arties from atherosclerotic plaques formation [72], whereas deficiency of adiponectin results in the higher incidence of atherosclerosis [73].

Mechanistically, many of the adiponectin benefits connected to its vasculoprotective are carried out via its ability to increases nitric oxide production through the activation of eNOS by AMPK-dependent manner [74,75]. It has also been proven that adiponectin prevents endothelial apoptosis through the AMPK mediated pathway [75]. Adiponectin supplementation reduces TNF-α mediated vascular cell adhesion molecule-1 and interleukin-8 by suppressing the nuclear factor kappa-b activation in endothelial cells [76,77]. Furthermore, cyclooxygenase-2 expression was increased by adiponectin treatment in cultured endothelial cells, and deletion of cyclooxygenase-2 inhibits adiponectin meditated growth in endothelial cell migration, differentiation, and survival [78]. Studies supports the idea that induction of cyclooxygenase-2 expression is mediated by sphingosine kinase-1 in cardiomyocytes by adiponectin [79]. Based on previous studies, it is evident that pressure overload or angiotensin II-induced cardiac hypertrophy, was inhibited through AMPK activation by adiponectin treatment in myocytes [56,69]; and, in animal models, adiponectin has been shown to be protective for systolic and diastolic dysfunction of myocardial infarction [80,81]. Thus, adiponectin might utilize AMPK pathway and cyclooxygenase-2 to improve endothelial function.

Although AdipoR1 and AdipoR2 are mainly involved in the metabolic action of adiponectin, some studies have investigated other receptors for adiponectin in heart [31,82]. Studies have shown that T-cadherin is a GPI-anchored adiponectin-binding protein involved in the cardioprotective action of adiponectin. T-cadherin is highly expressed in the vasculature, including endothelial cells [83], smooth muscle cells [84], and pericytes [85]. Studies demonstrate that ablation of T-cadherin abolishes adiponectin mediated cardioprotective effects in both short and long term cardiac hypertrophy as well as myocardial ischemia-reperfusion injury. This suggests that T-cadherin is a physiological adiponectin binding receptor that enables the association of adiponectin within the heart [34]. T-cadherin is also essential for the provascularization effects of adiponectin in mice [35]. Furthermore, hypoadiponectinemia was observed in T-cadherin-deficient mice, thus supporting the impairment of adiponectin recruiting in cardiovascular tissues of these mice [34]. Conversely, low T-cadherin tissue expression was observed in adiponectin-deficient mice, suggesting regulatory axis between T-cadherin and adiponectin [35].

### 4.3. Adipocyte/Adipose Tissue

Adipose tissue plays a central role in regulating whole-body energy and glucose homeostasis through its subtle functions at both organ and systemic levels [86]. Adipose tissue, which is primarily composed of adipocytes as well as pre-adipocytes, macrophages, endothelial cells, fibroblasts, and leucocytes, has been increasingly recognized as a major player of systemically metabolic regulation [87]. Adipose tissue acts as an endocrine organ and produces numerous bioactive factors such as adipokines that communicate with other organs and modulate a range of metabolic pathways. On the other hand, adipose tissue stores energy in the form of lipid and controls the lipid mobilization and distribution in the body [88].

Studies reported that 3T3L1 adipocytes showed high expression of adiponectin. Adiponectin, through its autocrine activity, helps in adipocytes cell differentiation. In adipocytes, C/EBPα, PPARγ, and sterol regulatory element-binding protein (SREBP)-1c are involved in promoting adipogenesis, and increasing lipid content and insulin directed glucose transport [89]. Transgene-mediated overexpression of adiponectin in ob/ob mice lead to morbid obesity due to decreased energy expenditure; however, there is marked improvement in glucose metabolism, accompanied by a reduction in macrophage numbers in adipose tissue and decreased expression of TNFα in fat pads. In addition, over expression of adiponectin in ob/ob mice showed improved vascularization and expansion of the subcutaneous fat pad in experimental animals. Collectively, chronic over expresson of adiponectin leads to massive increase in subcutaneous fat, and it protects against diet induced insulin resistance [90].

Overexpression of adiponectin protects against both the acute and the chronic effects of high fat diet HFD-induced lipotoxic effects of lipid accumulation, and, in mice, it increases the metabolic flexibility of the adipose tissue [91]. Adiponectin, through its receptor signaling, is also involved in the metabolic action of adipocyte/adipose tissue. Adiponectin levels in bloodstream play a key role in reflecting its metabolic action on adipocytes and adipose tissue. There is evidence showing that low adiponectin receptors are expressed in visceral adipocytes in adipose tissues of humans and rats, and decreased adiponectin receptor expression is detected in adipose tissues of insulin-resistant animals. These results indicate an impairment of adiponectin action in the insulin-resistant animals by low adiponectin receptor activity [92]. Additionally, it was reported that activation of PPARα with its agoinists in obese diabetic KKAy mice can stimulates the potency of adiponectin through upregulating the adiponectin and its receptor expressions in adipocyte/adipose tissue, which ultimately rescued these animals from obesity induced insulin resistance [57].

### 4.4. Liver

The liver plays a major role in blood glucose homeostasis by maintaining a balance between the uptake and storage of glucose via glycogenesis, and the release of glucose via glycogenolysis and gluconeogenesis [93]. Injection of recombinant adiponectin in both wild type and diabetic mouse models lowers serum glucose to near normal levels [45]. Intraperitoneal injection of HMW and LMW adiponectin lowers plasma glucose in healthy mice as well as mice with Type 1 Diabetes (T1D) and Type 2 Diabetes (T2D) [9]. In addition, high doses of adiponectin did not show hypoglycemic episodes in mice, which implies that the glucose lowering effect of adiponectin is primarily mediated by suppressing gluconeogensis or glycogenolysis. Short term infusion of adiponectin led to marked suppression of endogenous glucose production in conscious mice by suppressing glucose 6 phosphatase mRNA and phospho enol pyruvate carboxy kinase mRNA in the liver [94]. The insulin sensitizing action of adiponectin can also be mediated by up regulating PPARα, and its target genes include CD36, ACO, and UCP-2 in liver [5]. In addition, adiponectin supplementation has suppressed glucose output in primary rat hepatocytes [45]. Thiazolidinedione drugs restore the glycemic status by increasing the circulating levels of adiponectin in type 2 diabetic patients, adiponectin transgenic models, and knockout mouse models [91,95,96].

Adiponectin is not “insulin mimetic”. Adiponectin is effective in alleviating both alcohol and obesity associated liver abnormality, including hepatomegaly, steatosis, and the elevated levels of serum alanine aminotransferase. These therapeutic effects resulted partly from the ability of adiponectin to increase carnitine palmitoyltransferase I activity and enhance hepatic fatty acid oxidation, while decreased the activities of two key enzymes involved in fatty acid synthesis, including acetyl-CoA carboxylase and fatty acid synthase [97]. HMW and LMW adiponectin supplementation induces AMPK activation in in rat McArdle 7777 hepatoma cells [98]. Activated AMPK downregulates lipogenic genes and activates fat oxidative pathways [99]. Using noninvasive methods, it has been demonstrated that adiponectin levels in circulation were inversely correlated with liver fat content [100]. As patients with nonalcoholic steatohepatitis also have dysregulation of postprandial glucose and lipid homeostasis, we hypothesized that serum adiponectin levels in patients with nonalcoholic steatohepatitis would respond suboptimally to a mixed meal compared with the response in obese controls [101]. However, other studies have reported a lower staining of AdipoR2 in biopsies of patients with non-alcoholic steatohepatitis when compared to simple steatosis, which might be explained by post translational deregulation [102].

Mitochondrial dysfunction represents a central mechanism linking obesity with associated metabolic complications. In nonalcoholic steatoheaptitis, the liver mitochondria showed ultrastructural lesions and low activity of the respiratory chain complexes. This attenuation in the activity of the respiratory chain would results from an accumulation of reactive oxygen species (ROS) that oxidize stored fats to form lipid peroxidation products, which ultimately leads to steatohepatitis, necrosis, inflammation, and fibrosis [103]. Studies reported that mice lacking adiponectin resulted in high fat accumulation even under normal chow consumption [104]. In fact, adiponectin itself has been described as a PPAR-γ target gene [105]. This hepatic steatotic condition may be responsible for the malfunction of mitochondria. Supplementation of adiponectin rescues the mitochondrial functions, thereby lowering the mitochondrial lipid peroxidation products [82], which might represent a common mechanism underlying the multiple beneficial activities of this hormone in various obesity-related pathologies.

## 5. Can We Increase Circulatory Adiponectin Status?

Research demonstrated that pharmacological elevation of circulating adiponectin will become the promising therapeutic strategy to ameliorate obesity related diseases. The PPARγ agonists thiazolidinediones (TZDs), such as rosiglitazone and pioglitazone, have been shown to elevate the circulating levels of adiponectin in both animals and humans [5,106]. Troglitazone, is an oral antihyperglycemic agent, which increases adiponectin production in isolated human adipocytes [107]. Studies have shown that the insulin-sensitizing effects of TZDs are mediated at least in part by the acceleration of adiponectin production in adiponectin-null mice [108]. The phytochemicals astragaloside II and isoastragaloside I, isolated from the medicinal herb radix, were shown to alleviate hyperglycemia, and improve glucose tolerance and insulin sensitivity, presumably by augmenting adiponectin secretion [109]. Zataria multiflora improved insulin sensitivity and reduced glucose levels through adiponectin secretion, in fructose fed insulin resistant rats. This increase in adiponectin levels might be due to increase in PPARγ protein levels [110]. Recently, it was demonstrated that garlic extract, after 12 weeks, improves adiponectin levels in patients with metabolic syndrome [111]. The treatment of diabetic mice with cobalt, a heme oxygenase inducer, reduces visceral and subcutaneous obesity, and increases insulin sensitivity through the upregulation of adiponectin [112]. Studies have suggested that supplementation of l-cysteine increases the adiponectin secretion in adipoctyes [113] and manganese supplementation increases adiponectin secretion in adipocytes and in Zucker type 2 diabetic rats too [114]. In addition, studies indicate that adiponectin levels increase in healthy, nondiabetic volunteers treated with PPAR α/γ agonists [115]. In addition, it has been reported that temocapril decreases plasma glucose level and this action may be partly due to increase in adiponectin levels in patients with essential hypertension [116]. Thus, drugs targeting adiponectin synthesis would be helpful in treating obesity, diabetes, and cardiovascular disease.

In addition to pharmacotherapy, an acute bout of aerobic exercise results in a significant increase in plasma adiponectin levels in abdominally obese individuals. Adiponectin levels are inversely proportional to both total and abdominal fat mass [117,118]. Evidence suggests that an acute bout of vigorous aerobic exercise may result in a significant increase in plasma adiponectin levels in trained athletes [119,120]. In contrast, immediately following the cessation of exercise, adiponectin levels are reported to be unchanged [119] or even reduced [120] in trained individuals. Kriketos et al. [121] report that one week of aerobic training results in increased adiponectin levels in abdominally obese men. In addition, Numao and colleagues report that although circulating levels of both medium- and low-molecular weight adiponectin decreased immediately following a bout of vigorous aerobic exercise, the proportion of high molecular weight adiponectin was significantly increased [122]. Thus, vigorous whole-body exercise leads to an acute increase in plasma adiponectin levels in subjects.

Although the multiple benefits of thiazolidinediones are well established, there are a number of safety concerns and limitations that must be taken into consideration when selecting TZDs for the management of metabolic diseases. Troglitazone, the first agent of this class to be approved, was effective in controlling glycemia but was removed from the market because of serious liver toxicity. Heart failure is one of the most common side effects of TZDs [123] and as a result the FDA added a related “black box” warning. Pioglitazone treatment has been associated with increased risk of the development of edema in type 2 diabetic patients [124]. Another important safety issue of TZDs is their effect on bone metabolism [125]. Indeed, TZDs have been shown to decrease bone density and thus increase fracture risk [126]. Studies have shown that pioglitazone usage increased the risk of bladder cancer [127]. Moreover, treatment with TZDs is associated with an increase in body weight even though insulin resistance is reduced [128]. Glitazones have been associated with macular edema, a serious form of diabetic retinopathy that leads to vision loss [129].

## 6. Conclusions

Adiponectin is a fat-derived hormone that appears to play a crucial role in protecting against insulin resistance/diabetes and atherosclerosis. Decreased adiponectin levels are thought to play a central role in the development of type 2 diabetes, obesity and cardiovascular disease in humans. Research in humans and rodent models has consistently demonstrated the role of adiponectin as an important physiological regulator of insulin sensitivity, glucose, and lipid metabolism as well as cardiovascular homeostasis. Current studies conducted in human and animal models for obesity, diabetes, and atherosclerosis have reported on the potential role of adiponectin and adiponectin receptors for these metabolic diseases. As the production of endogenous adiponectin is impaired as an effect of obesity and related pathologies, a practical therapeutic approach is to use pharmacological or dietary interventions to restore the capacity of adipose tissue in secreting adiponectin. In the future, this unique strategy can probably serve as a potential novel and innovative therapeutic approach for treatment of the metabolic diseases.

## Figures and Tables

**Figure 1 ijms-18-01321-f001:**
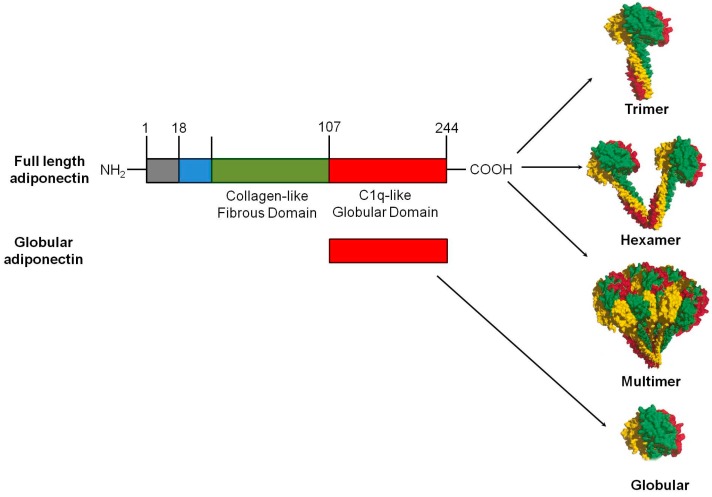
Domains and structure of adiponectin: Full-length adiponectin is composed of 244 amino acids, including a collagen-like fibrous domain at the N-terminus and a C1q-like globular domain at the C-terminus. In circulation, adiponectin forms low-molecular weight (LMW) homotrimers and hexamers, and high-molecular weight (HMW) multimers of 12–18 monomers. A smaller form of adiponectin that consists of globular domain also exists in plasma in negligible amounts. Each adiponectin subunit in the basic trimeric building block represented in a different color.

**Figure 2 ijms-18-01321-f002:**
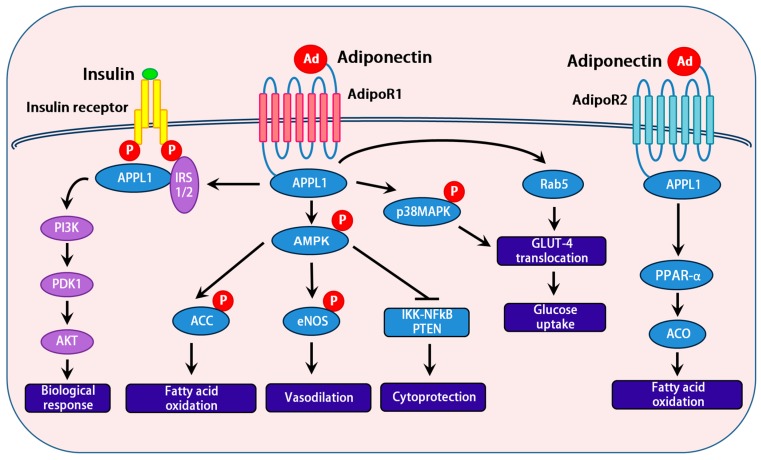
Schematic representation of adiponectin signal transduction implicating a cross talk with the insulin signaling pathway: Insulin and adiponectin interact with their respective receptors, which trigger a cascade of signaling events. Metabolic actions of the insulin are mainly carried out by PI3K/AKT pathway, resulting in increased protein synthesis, lipogenesis, glucose uptake and utilization, glycogen synthesis, and reduced lipolysis and gluconeogenesis. Interaction of adiponectin with its receptors (Adipo R1 and R2) results in the activation of multiple signaling pathways including IRS1/2, AMPK, and p38 MAPK. Activation of IRS1/2 by adiponectin signaling is a major mechanism by which adiponectin sensitizes insulin action in insulin responsive tissues.

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
