# Peer review of "Adiponectin, a Therapeutic Target for Obesity, Diabetes, and Endothelial Dysfunction"

_ijms, 2017, doi:10.3390/ijms18061321_

Round 1

Reviewer 1 Report

Adiponectin, a Therapeutic Target for Obesity, 2 Diabetes, and Endothelial Dysfunction by

Arunkumar E Achari and Sushil K Jain

This review article summarizes main functions of adiponectin. However, more recent papers like doi: 10.1038/nature21714, doi: 10.1016/j.molmet.2017.01.002, doi: 0.1080/14756366.2017.1284067 have to be discussed. Studies on AdipoR agonists have to be summarized in a separate paragraph.

Further, various spelling and grammatical errors have to be corrected.

Specific comments:

Abstract: „

 can also produce this adiponectin“ this should be deleted

“ Therefore, adiponectin replacement therapy“ it is not feasible to replace adiponectin and AdipoR agonists are the better option.

Introduction

“Adiponectin is an  endocrine factor released exclusively synthesized and then released from adipose tissue [5].” Please correct this sentence for grammatical errors.

3. Adiponectin: Biosynthesis, structure and downstream signaling “But, later it proven, from  different research groups, that adiponcetin expressed” please correct this sentence for spelling and grammatical errors

3.1.

“other tissues express reduced levels of adiponectin” reduced should be replaced by “low”

Fig. 1, there is one line without a number, please add amino acid position.

3.2.

“AdipoR1 is a high affinity receptor for globular adiopnectin” please correct spelling mistake.

“adipoR2 mainly“ please use upper case letter here and elsewhere.

“On the other  hand, adipoR2 mainly recognizes full length adiponectin and is predominantly expressed in the liver 133 [42]. “

42. Peter BJ, Kent HM, Mills IG, Vallis Y, Butler PJG, Evans PR and McMahon HT (2004) BAR domains as 520 sensors of membrane curvature: the amphiphysin BAR structure. Science 303:495-499.

Cited reference is wrong. Further initial studies using rodent tissues reveal preferential expression of AdipoR2 in the liver, in human tissues AdipoR1 and AdipoR2 mRNAs are most abundant in skeletal muscle and both are moderately expressed in the liver. Please see:

Yamauchi T, Kamon J, Ito Y, Tsuchida A, Yokomizo T, Kita S, Sugiyama T, Miyagishi M, Hara K, Tsunoda M, Murakami K, Ohteki T, Uchida S, Takekawa S, Waki H, Tsuno NH,Shibata Y, Terauchi Y, Froguel P, Tobe K, Koyasu S, Taira K, Kitamura T, Shimizu T, Nagai R, Kadowaki T. Cloning of adiponectin receptors that mediate antidiabetic metabolic effects. Nature 2003; 423: 762-769

3.4.

“APPL1-an adaptor protein  consisting of a PH pleckstrin homology domain, a phosphotyrosine binding domain, and a leucine 161 zipper motif-interacts directly with AdipoR1 and AdipoR2, and, thereby, mediates the actions of 162 adiponectin in the regulation of energy metabolism” this has been already explained in the paragraph above and has to be deleted.

“Activation of adiponectin is a key step in in mediating the most of the effects of adiponectin at cellular level.” Please correct.

uncouples proteins (UCPs);“ uncoupling proteins

Activation of ceramidas is a central pathway in adiponectin signaling and this should be discussed.

Figure 2 seems to be a summary of AdipoR1 signaling while AdipoR2 and PPARalpha are not shown.

Insulin signaling and AMPK signaling activate contrary biologic activities (energy storage and energy consumption). This has to be discussed in more detail.

4.1.

“By this reason, most of the adiponectin effects in skeletal muscle are carried out by globular form [31].”

Globular adiponectin is low abundant in serum, is this isoform indeed of physiological relevance?

“it activates in response to a numerous extracellular stimuli” “it is evident that APPL1 plays an important in adiponectin signaling. “ please correct grammatical errors

“peroxisome proliferator-activated receptor (PPAR) – α“ this abbreviation has been defined earlier.

4.2. “many of these adiponectin associated benefits connected to its  vasculoprotective are carried out via” “vascular cell adhesion to molecule-1 “please correct grammatical / spelling errors

In 3.2. authors write “Recent studies have identified that cadherin-deficient mice showed elevated plasma levels of adiponectin, especially HMW form of adiponectin [34].” And in 4.2. “Furthermore, hypoadiponectinemia was observed in T-cadherin-deficient mice, thus, supporting the  impairment of adiponectin recruiting in cardiovascular tissues of these mice [34].”

4.3.

“Adiponectin levels in the bloodstream and reflects metabolic  conditions in these cells/tissues.” please correct

4.4.

“In fact, adiponectin itself has been described as a PPAR- γ target gene [88].” This statement should be placed in a more appropriate context (paragraph 3?).

“heaptomegaly, statosis“ correct spelling errors.

“Patients with nonalcoholic steatohepatitis have an  uncontrolled postprandial glycemic and lipid homeostasis, and it is assumed that circulatory adiponectin levels in subjects with non-alcoholic steatohepatitis respond less to optimal levels in  meals compared with the response in obese controls [95].“ please rephrase.

“Studies reported that mice lacking adiponectin resulted in  high fat accumulation even under normal chow consumption [98].” Indeed, there are also studies showing different effects like 10.1016/j.yexmp.2012.03.008.

5.

meatbolic syndrome“

„Troglitazone, is an oral antihyperglycemic agent, which increases adiponectin production in isolated human adipocytes” this is an PPARgamma agonist and has to be listed above. Authors should indicate the problems of glitazone use.

Weight loss should be mentioned as one approach to increase systemic adiponectin

Author Response

This review article summarizes main functions of adiponectin. However, more recent papers like doi: 10.1038/nature21714, doi: 10.1016/j.molmet.2017.01.002, doi: 0.1080/14756366.2017.1284067 have to be discussed. Studies on AdipoR agonists have to be summarized in a separate paragraph. Further, various spelling and grammatical errors have to be corrected.

The revised manuscript has incorporated all of the comments and suggestions of the reviewers. Line by line reply to specific comments of each reviewer is given below:

Reviewer 1:

Specific comments:

Abstract:

1.     “can also produce this adiponectin” this should be deleted

Thank you for your suggestions. Now we have removed “can also produce this adiponectin” in the abstract.

2.     “Therefore, adiponectin replacement therapy“ it is not feasible to replace adiponectin and AdipoR agonists are the better option.

Thank you for your comments.  Studies have shown that recombinant adiponectin supplementation has been tested in different animal studies. Liu et al. 2013 [1] analyzed adiponectin administration in adiponectin knockout mice after high-fat diet feeding and found amelioration in metabolic profile (glucose handling, insulin signaling, triglycerides levels, and mitochondrial structure and function).  Moreover, Kondo et al. 2010 [2] demonstrated in mice and in pig models that adiponectin protects against ischemia/reperfusion injury through its ability to suppress inflammation, apoptosis, and oxidative stress. Ge et al., 2010 [3] reported that supplementation of recombinant adiponectin improve the fasting glucose levels and the tolerance to glucose in mice. Taking all these actions together, it is feasible that adiponectin is might be a better option for metabolic disease.

 1.     Liu, Y.; Turdi, S.; Park, T.; Morris, N.J.; Deshaies, Y.; Xu, A.; Sweeney, G. Adiponectin corrects high-fat diet-induced disturbances in muscle metabolomic profile and whole-body glucose homeostasis. Diabetes 2013, 62, 743-752.

2.     Kondo, K.; Shibata, R.; Unno, K.; Shimano, M.; Ishii, M.; Kito, T.; Shintani, S.; Walsh, K.; Ouchi, N.; Murohara, T. Impact of a single intracoronary administration of adiponectin on myocardial ischemia/reperfusion injury in a pig modelclinical perspective. Circulation: Cardiovascular Interventions 2010, 3, 166-173.

3.     Ge, H.; Xiong, Y.; Lemon, B.; Lee, K.J.; Tang, J.; Wang, P.; Weiszmann, J.; Hawkins, N.; Laudemann, J.; Min, X. Generation of novel long-acting globular adiponectin molecules. Journal of molecular biology 2010, 399, 113-119.

Introduction

3.     “Adiponectin is an endocrine factor released exclusively synthesized and then released from adipose tissue [5].” Please correct this sentence for grammatical errors.

As per your suggestions, we have now checked and corrected the sentence for grammatical errors (Page 4, Line 111- 112).

4.     3. Adiponectin: Biosynthesis, structure and downstream signaling “But, later it proven, from  different research groups, that adiponectin expressed” please correct this sentence for spelling and grammatical errors

As per your suggestions, we have now checked and corrected the sentence for spelling and grammatical errors (Page 4, Line 136- 137).

5.     “other tissues express reduced levels of adiponectin” reduced should be replaced by “low”

As per your suggestions, we have now changed the word to “low” (Page 4, Line 149).

6.     Fig. 1, there is one line without a number, please add amino acid position.

As per your suggestions, we have now included the amino acid position in the adiponectin structure description (Page 5, Line 151-153).

7.     3.2. “AdipoR1 is a high affinity receptor for globular adiponectin” please correct spelling mistake.

As per your suggestions, we have now corrected the spelling mistake (Page 6, Line 201).

8.     “adipoR2 mainly“ please use upper case letter here and elsewhere.

As per your suggestion, we have now used upper case letter for “adipoR2” and that has been followed in whole manuscript.

9.     “On the other hand, adipoR2 mainly recognizes full length adiponectin and is predominantly expressed in the liver  [42]. “

42. Peter BJ, Kent HM, Mills IG, Vallis Y, Butler PJG, Evans PR and McMahon HT (2004) BAR domains as 520 sensors of membrane curvature: the amphiphysin BAR structure. Science 303:495-499. Cited reference is wrong. Further initial studies using rodent tissues reveal preferential expression of AdipoR2 in the liver, in human tissues AdipoR1 and AdipoR2 mRNAs are most abundant in skeletal muscle and both are moderately expressed in the liver. Please see:

Yamauchi T, Kamon J, Ito Y, Tsuchida A, Yokomizo T, Kita S, Sugiyama T, Miyagishi M, Hara K, Tsunoda M, Murakami K, Ohteki T, Uchida S, Takekawa S, Waki H, Tsuno NH,Shibata Y, Terauchi Y, Froguel P, Tobe K, Koyasu S, Taira K, Kitamura T, Shimizu T, Nagai R, Kadowaki T. Cloning of adiponectin receptors that mediate antidiabetic metabolic effects. Nature 2003; 423: 762-769

As per your suggestions, we have now deleted the wrong reference and cited relevant literature to the text (Page 6, Line 225).

10. 3.4. “APPL1-an adaptor protein  consisting of a PH pleckstrin homology domain, a phosphotyrosine binding domain, and a leucine 161 zipper motif-interacts directly with AdipoR1 and AdipoR2, and, thereby, mediates the actions of 162 adiponectin in the regulation of energy metabolism” this has been already explained in the paragraph above and has to be deleted.

As per your suggestion, we have now deleted the above sentence. 

11. “Activation of adiponectin is a key step in in mediating the most of the effects of adiponectin at cellular level.” Please correct.

As per your suggestion, we have now corrected the sentence (Page 7, Line 247).

12. “uncouples proteins (UCPs);“ uncoupling proteins

As per your suggestion, we have now changed the word “uncouples proteins” to “uncoupling proteins” (Page 7, Line 253)

13.  Activation of ceramidas is a central pathway in adiponectin signaling and this should be discussed.

As per your suggestion, we have now discussed adiponectin and its relationship with ceramide in this manuscript (Page 8, Line 313-323).

14. Figure 2 seems to be a summary of AdipoR1 signaling while AdipoR2 and PPARalpha are not shown.

As per your suggestion, we have now shown the role of PPAR alpha in the summary diagram.

15. Insulin signaling and AMPK signaling activate contrary biologic activities (energy storage and energy consumption). This has to be discussed in more detail.

As per your suggestion we have now discussed insulin and AMPK signaling and its controversial biological activities in the manuscript (Page 7, Line 261-281).

16.  4.1. “By this reason, most of the adiponectin effects in skeletal muscle are carried out by globular form [31].”  Globular adiponectin is low abundant in serum, is this isoform indeed of physiological relevance?

Though, the concentration of globular adiponectin is low in serum, physiologically it has significant impact.  It is reported that globular adiponectin acutely increases glucose uptake and fat oxidation in human skeletal muscle [4]. In addition, globular adiponectin protected ob/ob mice from diabetes and prevents ApoE deficient mice from atherosclerosis [5].  Taking all these information together, globular form of adiponectin has great physiological relevance.

4.         Bruce, C.R.; Mertz, V.A.; Heigenhauser, G.J.F.; Dyck, D.J. The stimulatory effect of globular adiponectin on insulin-stimulated glucose uptake and fatty acid oxidation is impaired in skeletal muscle from obese subjects. Diabetes 2005, 54, 3154-3160.

5.         Yamauchi, T.; Kamon, J.; Waki, H.; Imai, Y.; Shimozawa, N.; Hioki, K.; Uchida, S.; Ito, Y.; Takakuwa, K.; Matsui, J. Globular adiponectin protected ob/ob mice from diabetes and apoe-deficient mice from atherosclerosis. Journal of Biological Chemistry 2003, 278, 2461-2468.

17. “it activates in response to a numerous extracellular stimuli” “it is evident that APPL1 plays an important in adiponectin signaling. “ please correct grammatical errors.

As per your suggestions, we have now corrected the above sentences (Page 8, Line 327). 

18. “peroxisome proliferator-activated receptor (PPAR) – α“ this abbreviation has been defined earlier.

As per your suggestion, we have now corrected the sentence (Page 8, Line 328).

19.  4.2. “many of these adiponectin associated benefits connected to its  vasculoprotective are carried out via” “vascular cell adhesion to molecule-1 “please correct grammatical / spelling errors.

As per your suggestion, we have now corrected the grammatical and spelling errors (Page 10, Line 378-379).

20. In 3.2. authors write “Recent studies have identified that cadherin-deficient mice showed elevated plasma levels of adiponectin, especially HMW form of adiponectin [34].” And in 4.2. “Furthermore, hypoadiponectinemia was observed in T-cadherin-deficient mice, thus, supporting the  impairment of adiponectin recruiting in cardiovascular tissues of these mice [34].”

Thank you for your comments.  As you noticed, it was a controversial observation.  We just want to report the readers that two different controversial observations have been found in cadherin-deficient mice.

21. 4.3. “Adiponectin levels in the bloodstream and reflects metabolic  conditions in these cells/tissues.” please correct

As per your suggestion, we have now corrected the sentence (Page 11, Line 436).

22.  4.4. “In fact, adiponectin itself has been described as a PPAR- γ target gene [88].” This statement should be placed in a more appropriate context (paragraph 3?).

As per your suggestion, we have now added the above sentence in paragraph 3 (Page 12, Line 489- 490).

23. “heaptomegaly, statosis“ correct spelling errors.

As per your suggestion, we have now corrected the spelling errors (Page 11, Line 466).

24. Patients with nonalcoholic steatohepatitis have an  uncontrolled postprandial glycemic and lipid homeostasis, and it is assumed that circulatory adiponectin levels in subjects with non-alcoholic steatohepatitis respond less to optimal levels in  meals compared with the response in obese controls [95].“ please rephrase.

As per your suggestion, we have now rephrased the sentence (Page 12, Line 475- 478).

25. Studies reported that mice lacking adiponectin resulted in high fat accumulation even under normal chow consumption [98].” Indeed, there are also studies showing different effects like 10.1016/j.yexmp.2012.03.008.

Thank you for your suggestion.  We do agree with your comments. 

26. “meatbolic syndrome“

As per your suggestion, we have now corrected the spelling errors (Page 10, Line 378-379).

27. Troglitazone, is an oral antihyperglycemic agent, which increases adiponectin production in isolated human adipocytes” this is an PPARgamma agonist and has to be listed above.

As per your suggestion, we have now included the above sentence in suitable place of this manuscript (Page 12, Line 500-501).

28. Authors should indicate the problems of glitazone use.

As per your suggestion, we have now included the problems of glitazone usage paragraph in the manuscript (Page 13, Line 535-548).

29. Weight loss should be mentioned as one approach to increase systemic adiponectin

As per your suggestion, we have now included the problems of glitazone usage paragraph in the manuscript (Page 13, Line 522-534).

Reviewer 2 Report

In this manuscript the authors performed a revision concerning the interference of adiponectin in obesity, diabetes and endothelial dysfunctions. The manuscript is well written, well-illustrated and deserves to be published. I suggest to perform several minor corrections:

Key words, all are included in the manuscript title;

I do not understand the presence of an index in this kind of manuscript;

The references were not organized according to the journal style guide.

Author Response

1.     Keywords, all are included in the manuscript title.

Journal guidelines say keywords should be under the abstract.  Hence, we have included keywords below the abstract.

2.     I do not understand the presence of an index in this kind of manuscript.

Since, this is a review paper, presence of an index briefly explain the content of the paper, which makes more understandable for the readers.

3.     The references were not organized according to the journal styles

As per your suggestion, we have now changed the references according to the journal styles.

Round 2

Reviewer 1 Report

1.       There are still grammatical errors:

Abstract: In addition to adipocytes, other cell types, such as  skeletal and cardiac myocytes and endothelial cells.

Contents: A brief overviews

2.: It stored energy in the

3.: Adiponectin, also known as Acrp30 [2] AdipoQ [11] GBP-28 [12] and apM1 [3] and  independently identified by four groups using different approaches [2,3,11,12].

3.4.: Activation of adiponectin is a key step in mediating the most of the effects of adiponectin at cellular level.

3.5. : in case of AMPK glucose can be entered in oxidative pathway 268 (catabolic) The 2 pathways appear to converge on the phosphorylation of AS160, which  further involved in GLUT-4 translocation [53,54].

4.3.: Adiponectin levels in the bloodstream plays a key role in reflecting its metabolic action in  adipocytes and adipose tissue.

4.4.: including heptomegaly, steatosis

5.: Adiponectin levels inversely proportional to both total and abdominal fat mass [117,118].

2.       „Studies on adiponectin-null  mice suggest that the insulin-sensitizing effects of TZDs are mediated at least in part by  induction of adiponectin production [108].” In the knock-outs adiponectin can not be induced.

Author Response

1. There are still grammatical errors:

Abstract: In addition to adipocytes, other cell types, such as skeletal and cardiac myocytes and endothelial cells.

This sentence has been corrected in the revised manuscript (page 2, Lines 35-37).

 Contents: A brief overviews

2.  It stored energy in the

This has been changed to: “Adipose tissue stored energy in the” (page 4, Lines 123-124).

3. Adiponectin, also known as Acrp30 [2] AdipoQ [11] GBP-28 [12] and apM1 [3] and independently identified by four groups using different approaches [2,3,11,12].

We have changed the sentences as “Adiponectin (also known as Acrp30 [2] AdipoQ [11] GBP-28 [12] and apM1 [3]) is a 244–amino acid protein secreted mainly by the adipose tissue. It was identified almost simultaneously by 4 different groups using different approaches [2,3,11,12]” (page 4, Lines 133-135).

3.4. Activation of adiponectin is a key step in mediating the most of the effects of adiponectin at cellular level.

This sentence has been corrected (page 7, Lines 247-248).

3.5. in case of AMPK glucose can be entered in oxidative pathway 268 (catabolic) The 2 pathways appear to converge on the phosphorylation of AS160, which  further involved in GLUT-4 translocation [53,54].

This sentence has been corrected in revised manuscript (page 7, Lines 275-276).

4.3. Adiponectin levels in the bloodstream plays a key role in reflecting its metabolic action in adipocytes and adipose tissue.

This sentence has been corrected in revised manuscript (page 11, Lines 437-438).

4.4.: including heptomegaly, steatosis

This sentence has been corrected in revised manuscript (age 12, Line 468).

5.: Adiponectin levels inversely proportional to both total and abdominal fat mass [117,118].

This sentence has been corrected in revised manuscript (page 13, Line 526).

2. Studies on adiponectin-null mice suggest that the insulin-sensitizing effects of TZDs are mediated at least in part by induction of adiponectin production [108].” In the knock-outs adiponectin cannot be induced.

This is corrected in the revised manuscript (page 12, Lines 503-505).

Authors are grateful to the Editor and the Reviewer for time spent and the valuable comments.